# A Differential Detection Method Based on a Linear Weak Measurement System

**DOI:** 10.3390/s19112473

**Published:** 2019-05-30

**Authors:** Nian Xiong, Tian Guan, Yang Xu, Lixuan Shi, Suyi Zhong, Xuesi Zhou, Yonghong He, Dongmei Li

**Affiliations:** 1College of Information Science and Technology, Jinan University, Guangzhou 510632, Guangdong, China; 2Institute of Optical Imaging and Sensing, Shenzhen Key Laboratory for Minimal Invasive Medical Technologies, Graduate School at Shenzhen, Tsinghua University, Shenzhen 518055, China; guantian@sz.tsinghua.edu.cn (T.G.); xuyang17@mails.tsinghua.edu.cn (Y.X.); slx16@mails.tsinghua.edu.cn (L.S.); zhongsy18@mails.tsinghua.edu.cn (S.Z.); zhouxs17@mails.tsinghua.edu.cn (X.Z.); 3School of Medicine, Tsinghua University, Beijing 100084, China; 4Department of Physics, Tsinghua University, Beijing 100084, China; 5Center for Optics & Optoelectronics Research, Collaborative Innovation Center for Information Technology in Biological and Medical Physics, College of Science, Zhejiang University of Technology, Hangzhou 310023, China; ldm20010@163.com

**Keywords:** weak measurement, differential detection, linear system

## Abstract

Self-reference detection is necessary and important to a biosensor. The linear weak measurement system based on total internal reflection has attracted widespread attention due to its high stability, label-free detection, and easy integration. In this paper, we propose a differential detection method based on the linear total internal reflection weak measurement system. We introduce the half-wave plate (HWP) to convert the H light and the V light to each other, thereby obtaining the difference in phase change of the optical path before and after the HWP. Experiments show that the system can not only achieve differential detection, but also has high stability. The linear differential weak measurement system proposed in this paper not only provides a new differential measurement method for real-time biosensors, but also enriches the types of weak measurement sensors.

## 1. Introduction

The principle of weak value amplification (WVA) was proposed in 1988 by Aharonov et al. [1] and realized by Ritchie et al. in 1991 for the first time in experiments [2]. It has gradually attracted widespread attention in society. In weak measurements, the measured parameters cause disturbances to the measuring device, providing a slight offset between the two eigenstates, which can be represented by two orthogonal polarization states in the optical system. With proper pre-selection and post-selection, this slight offset can be amplified and eventually be read out from the pointer received by the detector, which is the so-called weak value amplification. In recent years, weak measurements have shown great advantages in many high-precision measurements such as Goos–Hänchen and Imbert–Fedorov shifts [3,4], the photonic spin Hall effect [5], phase measurement [6,7], velocity measurement [8], temperature sensing [9], reflection angle of light beam [10], and optical rotation [11]. Differential detection methods are important in high precision measurements, also as a self-compensating switchable measurement method where dielectric properties are highly important. These methods compensate environment effect, voltage offset, frequency drift, and temperature influence, such as we can see in [12,13].

In 2010, Brunner et al. demonstrated the feasibility of weak measurement in the frequency domain, and the detection accuracy of the weak measurement in the frequency domain can be two to three orders of magnitude higher than the traditional interferometry [6]. The implementation of weak measurement in the frequency domain makes the weak measurement technology widely developed in the field of biomolecule detection. Our previous work has demonstrated that phase changes in the optical system induced by analytes or biochemical reactions in frequency domain weak measurements can cause a shift in the center wavelength of the output spectrum [14,15,16,17,18,19]. In addition, the combination of frequency domain weak measurement techniques and total internal reflection techniques provides significant advantages in phase sensitive biotransmissions on a single glass surface with a common path. At the same time, its easy integration in the structure of various devices provides great convenience for the application of weak measurement techniques to microscopes and some other instruments.

Self-testing is important in various biosensors. Implementing self-reference detection by differential means has always been a challenge for biosensors. Although some conventional optical systems, such as the difference interferometer reported in [20,21] and dual-mode surface-plasmon resonance (SPR) sensors [22], can be utilized as a differential measurement method, accurate detection of complex analyte concentrations, in these systems, may be subject to a number of disturbances, including non-target molecule binding and non-specific binding background parameter changes of the target molecule. However, because these instruments require either optimization of the angle of incidence and wavelength or precise control of the film thickness, the complexity of the preparation of these methods is greatly increased. A half-wave plate (HWP) can introduce a phase difference of π/2 between the polarized components paralleled to the fast axis and slow axis. In this paper, we convert the component of the H-(horizontal) and V-(vertical) polarized light, by tuning the fast axis angle to 45° to the vertical direction [23]. At the same time, the frequency domain weak measurement system based on total internal reflection (TIR) has the characteristics of simple structure and easy establishment, so it shows great advantages in differential measurement. In addition, the label-less, real-time detection characteristics of the weak measurement system also indicate its high practicability.

In this paper, we first proposed a linear common optical path weak measurement system based on TIR for differential detection. We introduced the HWP to convert the H light and the V light to each other, thereby obtaining the difference in phase change of the optical path before and after the HWP. In this work: (1) We verified by a Soleil–Babinet compensator (SBC) that the phase change before and after HWP in the system has an opposite effect on the center wavelength shift of the system. (2) We also simulated the refractive index change experiment by SBC, which proved that the system has extremely high resolution for the refractive index. The resolution was 2.34 × 10^−6^ Refractive Index Unit (RIU). (3) Because the system is realized by a linear common light path, it has extremely high stability. The system implements a more compact differential measurement method and provides great potential for self-reference detection in biosensor applications.

## 2. Theory

Weak measurement methods are based on the standard von Neumann measuring procedure [24,25]. We used an observing device to measure, indirectly, the state of the measuring system. The pointer state observable interacts with the observable of the measuring system. Hamiltonian of the measuring process can be then defined as
(1)$$\hat{H}=-g\left(t\right)\hat{P}\hat{A}$$
where g(t) is a normalized variable related to measuring time t as ∫g(t)dt=k. *k* is related to the time of interaction. The measuring system operator *P* describes the momentum of photons. Operator *A* represents the interaction between the measurement device and the system. We obtained the state of the system indirectly through the eigenstate of the readout observer *A*.

According to [26], the measurement process of a quantum system can be described as follows. Assuming that the object of measurement is *B* at time *t*_1_, the eigenvalue measured is *b_n_*; and *C* at time *t_2_*, the eigenvalue measured is *c_n_*. In time *t* (*t_1_ < t < t_2_*), the system can be represented by the wave functions of a bra $\langle \Psi_{post}|$ and a ket $|\Psi_{pre}\rangle$.
(2)$$|\Psi_{pre}\rangle =\underset{n}{\sum }\exp\left(-i\overset{t}{\underset{t_{1}}{\int }}Hd\tau \right)|B=b_{n}\rangle$$
(3)$$\langle \Psi_{post}|=\underset{n}{\sum }\langle C=c_{n}|\exp\left(-i\overset{t_{2}}{\underset{t}{\int }}Hd\tau \right)$$

$|\Psi_{pre}\rangle$ refers to the wave function evolving toward the future and $\langle \Psi_{post}|$ toward the past. This formula shows the probabilities of system state at time *t* when a system is pre-selected in the state $|\Psi_{pre}\rangle$ and post-selected in the state $\langle \Psi_{post}|$. Here the weak value of the observable *A*, playing a key role in amplification of the interaction, is defined as $A_{\omega }=\frac{\langle \Psi_{post}\left|A\right|\Psi_{pre}\rangle }{\langle \Psi_{post}{|\Psi }_{pre}\rangle }$.

We take the initial state of the measuring device to be the Gaussian:(4)$$|\Phi_{in}\rangle =\int d\tilde{P}\frac{1}{\sqrt{\Delta \tilde{P}}{\left(2\pi \right)}^{1/4}}\exp\left(-\frac{{\tilde{P}}_{i}^{2}}{4{\left(\Delta \tilde{P}\right)}^{2}}\right)|\tilde{P}\rangle .$$

For our frequency-domain system, the momentum is normalized as ${\tilde{P}}_{i}=\left(P_{i}-P_{0}\right)/P_{0}$, and $\Delta \tilde{P}=\Delta P/P_{i}$. Here, $P_{0}$ represents the center momentum of incident light, ΔP represents the uncertainty. $P_{i}$ can be regarded as the momentum of a single photon out of the incident photons. After coupling the system state with the measurement device state, the coupling state in the process can be expressed as
(5)$$\exp\left(-i\int \hat{H}dt\right)|\Psi_{pre}\rangle |\Phi_{in}\rangle =\sum_{n}a_{n}\int d\tilde{P}e^{i\tilde{P}a_{n}}\exp\left(-\frac{\tilde{P}}{4\Delta {\tilde{P}}^{2}}\right)|B=b_{n}\rangle |\tilde{P}\rangle .$$

We can therefore obtain the final state of the measuring device as
(6)$$|\Phi_{out}\rangle =\langle \Psi_{post}|e^{-i\int \hat{H}dt}|\Psi_{pre}\rangle |\Phi_{in}\rangle =\langle \Psi_{post}|e^{-i\int \hat{H}dt}|\Psi_{pre}\rangle \exp\left(-\frac{{\tilde{P}}^{2}}{4{\left(\Delta \tilde{P}\right)}^{2}}\right)|\tilde{P}\rangle =\;\langle \Psi_{post}{|\Psi }_{pre}\rangle \exp\left(i\tilde{P}\frac{\langle \Psi_{post}\left|A\right|\Psi_{pre}\rangle }{\langle \Psi_{post}{|\Psi }_{pre}\rangle }\right)\exp\left(-\frac{{\tilde{P}}^{2}}{4{\left(\Delta \tilde{P}\right)}^{2}}\right)|\tilde{P}\rangle +\langle \Psi_{post}{|\Psi }_{pre}\rangle \sum_{n=2}^{\infty }\frac{{\left(i\tilde{P}\right)}^{n}}{n!}\left[{\left(A^{n}\right)}_{\omega }-\;{\left(A_{\omega }\right)}^{n}\right]\exp\left(-\frac{{\tilde{P}}^{2}}{4{\left(\Delta \tilde{P}\right)}^{2}}\right)|\tilde{P}\rangle .$$

According to [1], the pointer state function is deduced as
(7)$$\exp\left[i\tilde{P}Re\left(A_{\omega }\right)\right]\exp\left(-\frac{{\left[\tilde{P}+2{\left(\Delta \tilde{P}\right)}^{2}Im\left(A_{\omega }\right)\right]}^{2}}{4{\left(\Delta \tilde{P}\right)}^{2}}\right).$$

This derivation should satisfy the first order approximation premise of ${\left(2\Delta \tilde{P}\right)}^{n}\frac{\Gamma \left(n/2\right)}{\left(n-1\right)!}\left|{\left(A^{n}\right)}_{\omega }-{\left(A_{\omega }\right)}^{n}\right|\ll 1$.

For our system in Figure 1, the system state is the polarization state, and the readout pointer state is the photon momentum state. The light source was a superluminescent laser diode (SLD), coupled by the collimating lens. We used two polarizers to pre-select and post-select the polarization state of light before and after the measurement process. Both of the two prisms were measured in this paper. There was an HWP in between the prisms to convert the polarization state of light in the horizontal and vertical direction. An SBC was placed after prism 2 to modulate the initial phase difference and adjust the system to the sensitive area. After the post-selection polarizer (P2), the light was collimated to a spectrometer (HR4000, Ocean Optics).

The polarizer 1 (P1) and 2 (P2) prepared the incident and final state, respectively, as
(8)$$|\Psi_{in}\rangle =\cos\alpha |H\rangle +\sin\alpha |V\rangle$$
(9)$$\langle \Psi_{out}|=\langle H|\cos\beta -\langle V|\sin\beta .$$

Here, H refers to the horizontal polarized light and V refers to the vertical polarized light. The angle of polarizer 1 was in α to the vertical direction and the angle of polarizer 2 was in β to the vertical direction.

A refractive index-dependent phase difference Δ between p and s polarizations was added by each prism. The phase difference from two prisms were $\Delta_{1}$ and $\Delta_{2}$, respectively. Because the interaction time is related to the total phase difference, we can obtain *k* as $k=\Delta_{1}+\Delta_{2}$. According to Fresnel’s formula, Δ can be expressed as
(10)$$\Delta =2\tan^{-1}\frac{\sqrt{{\left(n_{0}/n_{s}\right)}^{2}\sin^{2}\theta -1}}{n_{s}\sin\theta \tan\theta /n_{0}}.$$

Here, $n_{0}$ refers to the refractive index of the prism, $n_{s}$ refers to the refractive index of the sample on the reflection surface of the prism. Under the first order approximation ($n_{s}=n+dn,dn\ll 1$), dΔ has an approximate linear relationship to dn, which can be expressed by the following formula:(11)$$d\Delta =\frac{2n\sin\theta \tan\theta }{n_{0}^{2}\left[\sin^{2}\theta {\left(n_{0}/n\right)}^{2}\left(1+\tan^{2}\theta \right)-1\right]{\left(\sin^{2}\theta {\left(n_{0}/n\right)}^{2}-1\right)}^{1/2}}dn=\eta dn.$$

The relationship between Δ and *n* is given in Figure 2. In the measurement range, Δ and n keep in good linearity. Therefore, we chose the method of modulating the phase, by an SBC, to simulate the effect of changing the refractive index of the prism. For SBC, the resulting phase difference is as follows:(12)$$\delta =\frac{2\pi }{\lambda }\left(d_{1}-d_{2}\right)\left(n_{o}-n_{e}\right),$$
where $d_{1}$ and $d_{2}$ refer to the thicknesses of the birefiringent crystal plates of the compensator. $n_{o}$ and $n_{e}$ denote the refractive indexes for the extraordinary and ordinary components, respectively. We adjusted the fast axis of the SBC to the vertical direction, which caused a phase difference of δ between the H light and V light. The change of refractive index n can be simulated by adjusting δ, and the experiment can be carried out either by changing the RIU on prism surface or by adjusting the phase difference to simulate the RIU change. 

After the first prism, the system state yields out as
(13)$$\sin\alpha e^{i\Delta_{1}}|H\rangle +\cos\alpha e^{-i\Delta_{1}}|V\rangle .$$

After that, the light passed through an HWP plate. The plate was placed 45° to the vertical direction. The operator of HWP can be regarded as
(14)$$HWP=e^{i\pi /2}|A\rangle \langle A|+e^{-i\pi /2}|B\rangle \langle B|$$
where $|A\rangle =\frac{\sqrt{2}}{2}|H\rangle +\frac{\sqrt{2}}{2}|V\rangle$ and $|B\rangle =\frac{\sqrt{2}}{2}|H\rangle -\frac{\sqrt{2}}{2}|V\rangle$ refers to the fast axis and slow axis, respectively. Therefore, after the HWP, the polarization state of light can be derived as
(15)$$\cos\alpha e^{-i\Delta_{1}}|H\rangle +\sin\alpha e^{i\Delta_{1}]}|V\rangle .$$

We split the system in the second prism; the split was chosen as $x\lambda_{0}$ to the end of the evanescent wave, $\lambda_{0}$. 

*x* refers to the phase difference of the split part. In this condition, the pre- and post-selection states can be respectively derived as
(16)$$|\Psi_{pre}\rangle =\cos\alpha e^{i[\left(\Delta_{2}-x\right)-\Delta_{1}}|H\rangle +\sin\alpha e^{-i\left[\left(\Delta_{2}-x\right)-\Delta_{1}\right]}|V\rangle$$
(17)$$|\Psi_{pre}\rangle =-\langle V|\sin\beta e^{-ix}+\langle H|\cos\beta e^{ix}.$$

The measurement can also be regarded in the first prism. If we split the evanescent wave by $x\lambda_{0}$ to the end, the selection states are
(18)$$|\Psi_{pre}\rangle =\cos\alpha e^{i\left(\Delta_{1}-x\right)}|H\rangle +\sin\alpha e^{-i\left(\Delta_{1}-x\right)}|V\rangle$$
(19)$$|\Psi_{pre}\rangle =-\langle H|\sin\beta e^{i\left(-\Delta_{2}+x\right)}+\langle V|\cos\beta e^{-i\left(-\Delta_{2}+x\right)}.$$

Weak value $A_{\omega }$ can be derived to have no relationship to *x*, as
(20)$$A_{\omega }\equiv \frac{\langle \Psi_{post}\left|A\right|\Psi_{pre}\rangle }{\langle \Psi_{post}{|\Psi }_{pre}\rangle }=\frac{\cos\alpha \sin\beta e^{i\left(\Delta_{2}-\Delta_{1}\right)}+\sin\alpha \cos\beta e^{-i\left(\Delta_{2}-\Delta_{1}\right)}}{\cos\alpha \sin\beta e^{i\left(\Delta_{2}-\Delta_{1}\right)}-\sin\alpha \cos\beta e^{-i\left(\Delta_{2}-\Delta_{1}\right)}}.$$

To simplify the formulation, we obtained
$$A_{\omega }=\frac{1+\gamma e^{iD}}{1-\gamma e^{-iD}}\;\mathrm{and}\;{\mathrm{ImA}}_{\omega }\approx \frac{2\gamma \sin D}{1+\gamma^{2}-2\gamma \cos D}$$
where γ=cotαtanβ refers to the angle of the pre- and post-selected polarizer, and $D=\Delta_{2}-\Delta_{1}$ refers to the phase difference to be measured. According to [2], the shift of momentum can be derived with the weak value as $\delta \tilde{P}=2k{\left(\Delta \tilde{P}\right)}^{2}Im\left(A_{\omega }\right)$. The wavelength of light is inversely proportional to its momentum, derived as λ=ℏ/P. We can therefore derive the wavelength shift to be $\delta \lambda =\frac{2k{\left(\Delta \lambda \right)}^{2}Im\left(A_{\omega }\right)}{\lambda_{0}^{2}}=\frac{8\pi k{\left(\Delta \lambda \right)}^{2}\gamma \sin D}{\lambda_{0}^{2}\left(1+\gamma^{2}-2\gamma \cos D\right)}$. $\lambda_{0}$ refers to the center wavelength of the incident gauss light, and Δλ refers to the uncertainty of the wavelength of each photon. Thus, we derived the total phase difference of two prism $D=\Delta_{2}-\Delta_{1}$ by measuring the center wavelength shift in the spectrum. Figure 2a shows the relationship of center wavelength to the phase difference. In this paper, we selected the sensitive measurement area marked by blue arrows, and Figure 2(b1–b5) represents the spectral diagrams of five marked points.

## 3. Experiments

As has been explained in the theory, the frequency domain weak measurement system has a highly sensitive linear detection area where the readout spectrum is bimodal. This area is commonly used to detect areas of weak measurement systems, so we adjusted the system inside the bimodal area in the experiments below.

In order to verify the feasibility of the scoring system, as shown in Figure 3, we added an SBC1 in front of prism 1, based on the system of Figure 1, to simulate the change of the refractive index of the sample on the surface of prism 1, by modulating the phase difference of the system. The refractive index change of the sample on the surface of prism 2 was modulated by the SBC2 in the figure.
First, the SBC1 was modulated by a step of 0.005 mm (0.0027 rad of phase difference) each time, with the SBC2 unchanged in our system. For each step, we recorded the real-time the center wavelength shift of the spectrum acquired by the HR4000 spectrometer. As shown in Figure 4 by the blue line, as the phase difference of SBC1 increased, the system center wavelength shift gradually increased in a linearly way to the positive side. The slope was 1013.8 nm/rad.Similarly, SBC2 was adjusted in the same way as in 1, with SBC1 unchanged. For each step, the center wavelength shift of the spectrum was recorded by the HR4000 spectrometer. As shown in Figure 4 by the red line, as the phase difference of SBC2 increased, the system center wavelength shift gradually decreased in a linearly way to the negative side, by a slope of −1070.4 nm/rad, as opposed to the system center wavelength in 1 with the SBC scale change.Finally, to evaluate the differentiating effect, we adjusted SCB1 and SBC2 in the same direction simultaneously and kept adjusting the same step as in 1 and 2. After each adjustment of the two SBCs, we recorded the central wavelength shift of the spectrum acquired by the spectrometer. As shown in Figure 4 by the green line, the center wavelength of the system remains essentially the same (slope of 23.1 nm/rad) when the two SBCs were modulated in this simultaneous way.

From this we demonstrate that, by introducing the HWP, phase difference before and after the HWP has an opposite effect on the center wavelength shift of the system. Therefore, it can be deduced with Equation (20) that the refractive index change of the samples on the surface of two prisms has an opposite effect on the center wavelength shift of the system. In this way, we can verify that this differential measurement system can achieve differential detection.

The differential measurement was evaluated by changing the refractive index of the sample. According to the theory, we have proved that the refractive index of the sample can be simulated by adjusting the phase difference. The system structure diagram is shown in Figure 5a. Figure 5b shows the relationship between the refractive index of the sample and the phase difference. The linear-fitted equation is $\Delta \phi =2.542\Delta n,{\;r}^{2}=1$. Δn denotes the refractive index of the sample and Δϕ denotes the phase difference that should be chosen. We took $3.12\times 10^{-4}\;\mathrm{RIU}$ as one step to evaluate the center wavelength shift to the refractive index of sample A. The error bar chart is shown in Figure 5b. The measured data were averaged by every 100 data points. The sensitivity of our system can be derived by the slope of the linear-fitted function, which is $k_{0}=\delta \lambda /\delta n=5231.9\;\mathrm{nm}/\mathrm{RIU}$. Here, δλ/δn refers to the relationship between the central wavelength shift δλ and the change of refractive index δn of the sample. The resolution of center wavelength shift was derived as three times the mean standard deviation of each refractive index, which turns out to be $3{\bar{\sigma }}_{\delta \lambda }=0.0012\;\mathrm{nm}$. Therefore, we obtained the resolution of the refractive index in this system as $\sigma =3{\bar{\sigma }}_{\delta \lambda }/k_{0}=2.34\times 10^{-6}\;\mathrm{RIU}$. This value is better than the measurement results of our recent work on the common optical path [27].

In order to verify the feasibility of our proposed differential system in the field of biological detection, we conducted the following experiments. First, we passed deionized water (DW) into the flow path of prism A and prism B, and then adjusted the system to the bimodal position. We recorded the center wavelength of the bimodal spectrum of the system at this time and used this as a benchmark. As shown in Figure 6, we kept the material in the flow path of the prism B unchanged, and sequentially passed deionized water, 10 g/L glucose (Glu) solution, 2 g/L sodium chloride (NaCl) solution, and a mixed solution of the two substances (10 g/L glucose and 2 g/L sodium chloride) in prism A. The offset of the center wavelength of the system after each solution was recorded. Then, we replaced the material in the flow path of prism B with 10 g/L glucose solution, 2 g/L sodium chloride solution, and a mixed solution of the two substances (10 g/L glucose and 2 g/L sodium chloride), and then repeated the above experiment. We recorded the center wavelength of the bimodal spectrum of the system. Table 1 shows the offset values of the system center wavelength relative to the reference for each set of experiments.

From Figure 6 and Table 1, we can see that our proposed system can achieve differential detection of real samples.

It can be seen from Figure 6 and Figure 7 and Table 1 that the system can realize differential detection and has high refractive index resolution. In addition, because our system uses a linear common optical path structure, the light path does not need to split, so the system has higher stability than the weak measurement system based on the Mach–Zehnder [28] and Sagnac interferometer in the previous work. As shown in Figure 7, the stability of our test system is 6 h. In the experiment, we adjusted the system spectrum in the bimodal area, and continuously collected the output spectrum with the HR4000 spectrometer and recorded the center wavelength shift using a self-made program. In order to quantitatively analyze the stability of the system, we calculated the standard deviation of the experimental data in one hour, and the standard deviation was calculated as 0.00077 nm. The measured data were averaged by every 100 data points. The standard deviation from our previous work was less than the standard deviation of the weak measurement system based on the Mach–Zehnder (0.032 nm in 20 s) and Sagnac interferometer (0.0178 nm in 1 h). Therefore, the system in this paper has higher stability.

## 4. Discussion

In the weak measurement system, since the pre- and post-selected polarization states are almost orthogonal, the optical weak measurement sensor system suffers a certain loss of light intensity, causing the signal to be weak, and reducing the signal to noise ratio (SNR). Thus, the SNR can be increased by using a larger power light source or increasing the spectrometer integration time.At the same time, environmental factors also have a great influence on the weak measurement system. Factors such as temperature and vibration may cause errors in the experimental results. The external environment is required to be relatively stable to carry out a precise measurement. In this experiment, we controlled the ambient temperature to 25 degrees Celsius (0.1 degree Celsius) and built the system on an airborne optical platform.

## 5. Conclusions

In summary, in this paper we proposed a differential weak value amplification (WVA) measurement method that can be based on a linear common-path weak measurement system. A two-channel design using an HWP connection (for converting H and V polarizations to each other) enables differential measurements. Through the system, the phase difference before and after the HWP can be evaluated by the center wavelength shift in the output spectrum, and the differential detection of biomolecules can be realized by the TIR structure. We demonstrated the feasibility of implementing differential measurements based on the TIR weak measurement system by comparing the refractive index of samples before and after HWP, by modulating SBC, on the center wavelength shift of the system. At the same time, because of the linear common-path design, this system has higher stability than the Mach–Zehnder weak measurement system. In addition, since the phase difference between the H and V polarizations in the structure of the TIR was caused by the sample, which contacts the prism interface through total internal reflection, the light does not actually pass through the sample, so this measurement method is promising in a broad range of organism applications. The linear differential weak measurement system proposed in this paper not only provides a new differential measurement method for real-time biosensors, but also enriches the types of weak measurement sensors.

## Figures and Tables

**Figure 1 sensors-19-02473-f001:**
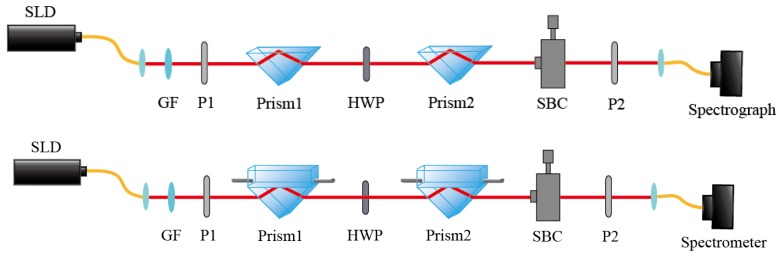
System structure diagram. SLD: superluminescent laser diode; GF: Gaussian filter; HWP: half-wave plate; SBC: Soleil–Babinet compensator.

**Figure 2 sensors-19-02473-f002:**
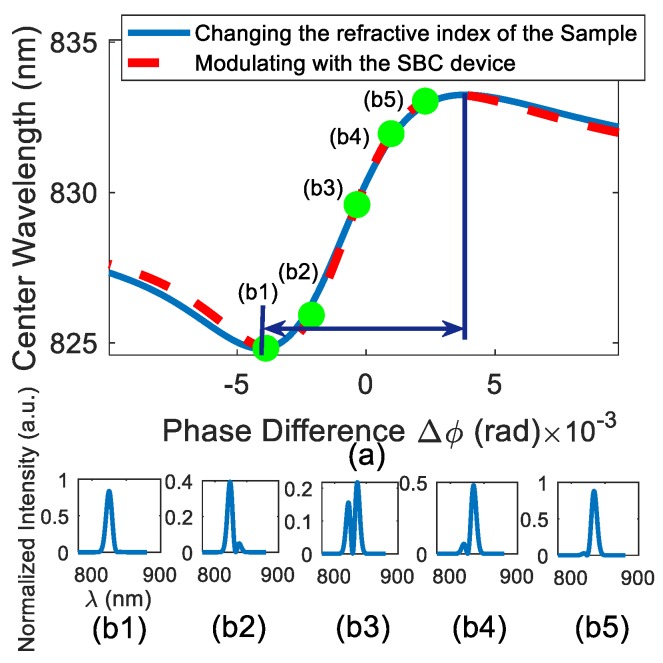
(**a**) The phase difference in the system affects the center wavelength of the system (the blue line in the figure is the phase difference in the system that changes by adjusting the SBC, and the red line in the figure changes the phase difference in the system by changing the refractive index of the total internal reflection surface of the prism). (**b**) b1–b5 are the spectra corresponding to the marked points in Figure 2a.

**Figure 3 sensors-19-02473-f003:**
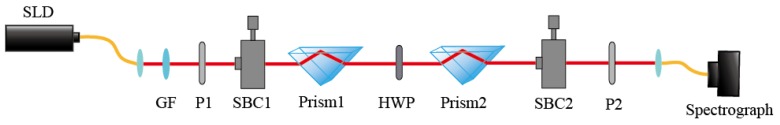
System structure diagram for verifying the differential feasibility test of the weak measurement system.

**Figure 4 sensors-19-02473-f004:**
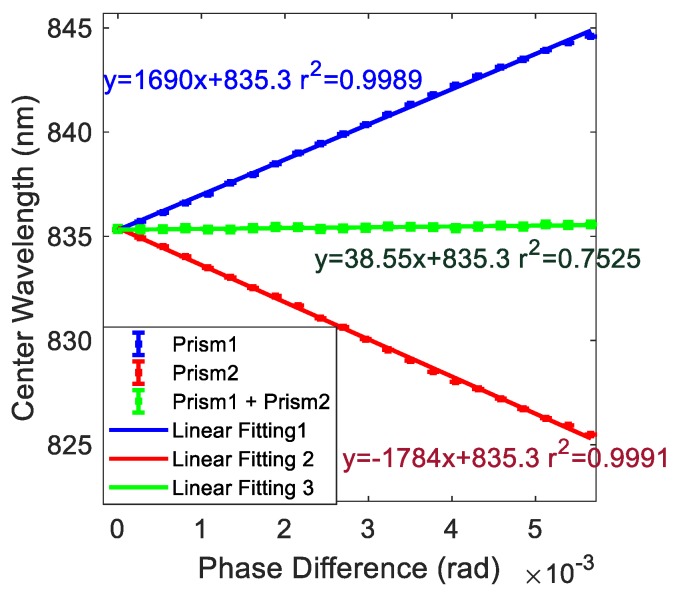
A schematic diagram of the system center wavelength offset as a function of phase difference at different locations within the system.

**Figure 5 sensors-19-02473-f005:**
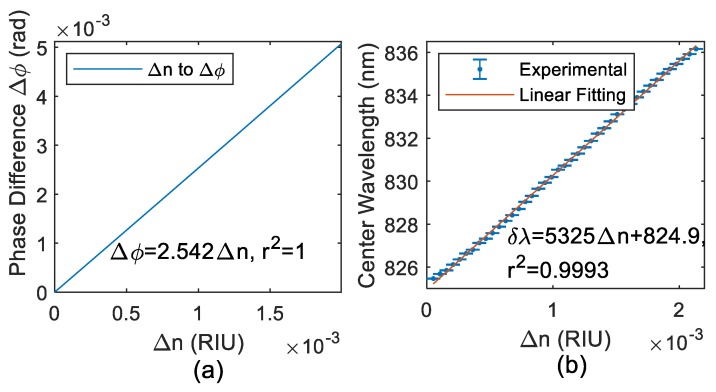
(**a**) In the system of this paper, the relationship between the refractive index change of the sample and the phase change of the system. (**b**) Relationship between the center wavelength shift of the system and the refractive index of the sample.

**Figure 6 sensors-19-02473-f006:**
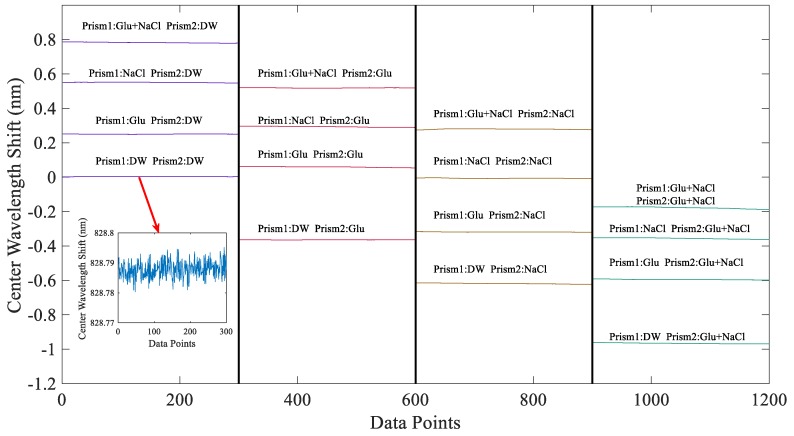
When a different solution is introduced into the flow paths of prism 1 and prism 2, the center wavelength of the system shifts. The inset is the central wavelength at which the deionized water is simultaneously introduced into the flow paths of prism 1 and prism 2.

**Figure 7 sensors-19-02473-f007:**
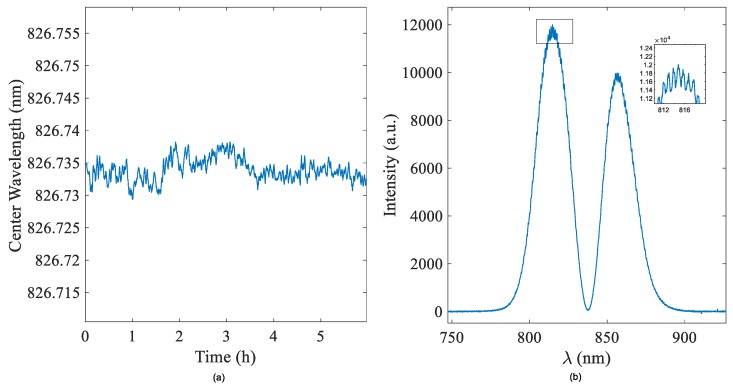
(**a**) System stability within 6 h. We used distilled water as the solution in two prisms; the temperature was 25 ± 0.1 degrees Celsius. (**b**) The spectrum. The inset shows the noise uncertainty of the spectrometer.

**Table 1 sensors-19-02473-t001:** The value of the system center wavelength shift amount when a different solution is passed through the flow path of prism 1 and prism 2. DW: deionized water.

	Prism 2	DW	Glu	NaCl	Glu + NaCl
Prism 1					
DW		0.0029	0.2503	0.5506	0.7823
Glu		−0.3643	0.0590	0.2932	0.5188
NaCl		−0.6184	−0.3187	−0.0063	0.2791
Glu + NaCl		−0.9659	−0.5950	−0.3559	−0.1774

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
