# Peer review of "A Differential Detection Method Based on a Linear Weak Measurement System"

_sensors, 2019, doi:10.3390/s19112473_

Round 1
Reviewer 1 Report
The observations were taken account.
In my opinion this paper could be accepted.
Author Response
Reviewer 1
Thank you for your recognition of our work.
Reviewer 2 Report
Comments to the authors:
The manuscript is well written but it needs some corrections.
Page 1: 1. Introduction: Line 39: Authors should include: “A differential detection methods are important in high precision measurement also as a self compensating switchable measurement methods where dielectric properties are highly important. These methods compensate environment effect, voltage offset, frequency drift, and temperature influence such as we can see in ref.: “
Matko V. and Milanović M., Temperature-Compensated Capacitance-Frequency Converter with High Resolution, Sens. Actuators A, 220, 2014, 262-269, doi: 10.1016/j.sna.2014.09.022.
http://www.sciencedirect.com/science/article/pii/S0924424714004178
Matko V., Next generation AT-cut quartz crystal sensing devices. Sensors, 2011, vol. 5, 11, 4474-4482,
doi: 10.3390/s110504474.
Authors should include the text and references above into the manuscript.
Pages 3, 5: Fig. 1 and Fig 2 are not highlighted (explanation) in the text (on Fig. 1 we can see …….).
Page 9:
Line 246: It can be seen from 1, 2 ??? ….. Explain!!
Line 245: Figure 7, System ……… - needs more comment!!
Line 251: Could authors get more data about measurement uncertainty of HR4000 spectrometer.
Line 256: … is less than the standard deviation (How much?)…….
Line 265: Why the environment temperature must be stable if you use differential measurement method?
Author Response
Reviewer 2
The manuscript is well written but it needs some corrections.
Page 1: 1. Introduction: Line 39: Authors should include: “A differential detection methods are important in high precision measurement also as a self compensating switchable measurement methods where dielectric properties are highly important. These methods compensate environment effect, voltage offset, frequency drift, and temperature influence such as we can see in ref.: “
Matko V. and Milanović M., Temperature-Compensated Capacitance-Frequency Converter with High Resolution, Sens. Actuators A, 220, 2014, 262-269, doi: 10.1016/j.sna.2014.09.022.
http://www.sciencedirect.com/science/article/pii/S0924424714004178
Matko V., Next generation AT-cut quartz crystal sensing devices. Sensors, 2011, vol. 5, 11, 4474-4482,
doi: 10.3390/s110504474.
Authors should include the text and references above into the manuscript.
Thank you for your suggestion. We have added the above content and references in the text. Line 40 to Line 43.
Pages 3, 5: Fig. 1 and Fig 2 are not highlighted (explanation) in the text (on Fig. 1 we can see …….).
Thank you for your correction. We have added relevant content in the text. Fig. 1 in Line 112. Fig 2 in Line 116 and Line 117.
Page 9:
Line 246: It can be seen from 1, 2 ??? ….. Explain!!
Thank you for your correction. We have corrected it in the text. Line 250.
Line 245: Figure 7, System ……… - needs more comment!!
Thank you for your suggestion. We have added relevant content in the text. Line 247 and Line248.
Line 251: Could authors get more data about measurement uncertainty of HR4000 spectrometer.
Thank you for the suggestion. We have added the spectrum of HR4000 in Fig. 7 and shown the noise uncertainty in the inset.
Line 256: … is less than the standard deviation (How much?)…….
Thank you for your suggestion. We have added relevant content in the text. Line 260.
Line 265: Why the environment temperature must be stable if you use differential measurement method?
Thank you for your question. The refractive index of the prism itself changes due to temperature changes, and errors are caused if temperature stability is not guaranteed.
This manuscript is a resubmission of an earlier submission. The following is a list of the peer review reports and author responses from that submission.
Round 1
Reviewer 1 Report
This is an interesting manuscript proposing a differential detection method based on the linear total internal reflection weak measurement system. The manuscript is well-written and the content fits to the scope of the journal. I suggest its acceptance in its present form. Minor comments:
Line 33: replace AAV by Aharonov et al.
Line 34: replace R by Ritchie et al.
Reviewer 2 Report
See attached.

Reviewer 3 Report
In this paper is reported a differential detection method based on the linear total internal reflection weak measurement system. The authors claim that linear differential weak measurement system proposed provides a new differential measurement method for the real-time biosensor.
In my opinion, the authors must improve this paper taking account the next:
The authors mention as previous work only reference 12. However, there are other previous works that are not discuss in this paper:
Li, Dongmei and Guan, Tian and He, Yonghong and He, Qinghua and Zhang, Yilong and Wang, Xiangnan and Shen, Zhiyuan and Yang, Yuxuan and Qiao, Zhen and Ji, Yanhong, “A differential weak measurement system based on Sagnac interferometer for self-referencing biomolecule detection,” Journal of Physics D: Applied Physics 50, 49LT01, (2017).
Yang Xu, Lixuan Shi, Tian Guan, Dongmei Li, Yuxuan Yang, Xiangnan Wang, Zhangyan Li, Luyuan Xie, Xuesi Zhou, Yonghong He, and Wenyue Xie, "Optimization of a quantum weak measurement system with digital filtering technology," Appl. Opt. 57, 7956-7966 (2018).
Yang Xu, Lixuan Shi, Tian Guan, Cuixia Guo, Dongmei Li, Yuxuan Yang, Xiangnan Wang, Luyuan Xie, Yonghong He, and Wenyue Xie, "Optimization of a quantum weak measurement system with its working areas," Opt. Express 26, 21119-21131 (2018).
Y.-J. Zhang, L.-X. Shi, Y. Xu, X. Zheng, J.-W. Li, Q. Wu, S.-X. Li, and Y.-H. He, "Optical quantum weak measurement coupled with UV spectrophotometry for sensitively and non-separatedly detecting enantiopurity," Opt. Express 27, 9330-9342 (2019).
Guan, Tian and Yang, Yuxuan and Zhang, Qianwen and He, Yonghong and Xu, Naihan and Li, Dongmei and Shi, Lixuan and Xu, Yang and Wang, Xiangnan, “Label-free and Non-destruction Determination of Single- and Double-Strand DNA based on Quantum Weak Measurement,” Scientific Reports 9, 1891-1899 (2019).
I consider important to mention and discuss such works regarding the method proposed in this paper.
What is the new contribution and how improves the previous results obtained for biosensors?
2. Due to the authors claim that the measurement system proposed provides a new differential measurement method for the real-time biosensor, I suggest to include a biological sample in the experiment.